# Troxerutin flavonoid has neuroprotective properties and increases neurite outgrowth and migration of neural stem cells from the subventricular zone

Muhammad Irfan Masood[1,2,3]*, Karl Herbert Schäfer[2]*, Mahrukh Naseem[4], Maximilian Weyland[2], Peter Meiser[5]

1 Division of Bioorganic Chemistry, School of Pharmacy, Saarland University, Saarbrücken, Germany, 2 ENS Group, University of Applied Sciences Kaiserslautern, Zweibrücken, Germany, 3 Institute of Pharmaceutical Sciences, University of Veterinary and Animal Sciences, Lahore, Pakistan, 4 Department of Zoology, University of Balochistan, Quetta, Pakistan, 5 Medical Scientific Department GM, URSAPHARM Arzneimittel GmbH, Saarbrücken, Germany

* karlherbert.schaefer@hs-kl.de (KHS); Irfan_masood_79@yahoo.com, mohammed.irfan.masood@gmail.com (MIM)

**Data Availability Statement:** All relevant data are within the manuscript and its Supporting Information files.

## Abstract

Troxerutin (TRX) is a water-soluble flavonoid which occurs commonly in the edible plants. Recent studies state that TRX improves the functionality of the nervous system and neutralizes Amyloid-ß induced neuronal toxicity. In this study, an *in vitro* assay based upon Neural stem cell (NSCs) isolated from the subventricular zone of the postnatal balb/c mice was established to explore the impact of TRX on individual neurogenesis processes in general and neuroprotective effect against ß-amyloid 1–42 (Aß42) induced inhibition in differentiation in particular. NSCs were identified exploiting immunostaining of the NSCs markers. Neurosphere clonogenic assay and BrdU/Ki67 immunostaining were employed to unravel the impact of TRX on proliferation. Differentiation experiments were carried out for a time span lasting from 48 h to 7 days utilizing ß-tubulin III and GFAP as neuronal and astrocyte marker respectively. Protective effects of TRX on Aß42 induced depression of NSCs differentiation were determined after 48 h of application. A neurosphere migration assay was carried out for 24 h in the presence and absence of TRX. Interestingly, TRX enhanced neuronal differentiation of NSCs in a dose-dependent manner after 48 h and 7 days of incubation and significantly enhanced neurite growth. A higher concentration of TRX also neutralized the inhibitory effects of Aß42 on neurite outgrowth and length after 48 h of incubation. TRX significantly stimulated cell migration. Overall, TRX not only promoted NSCs differentiation and migration but also neutralized the inhibitory effects of Aß42 on NSCs. TRX, therefore, offers an interesting lead structure from the perspective of drug design especially to promote neurogenesis in neurological disorders i.e. Alzheimer's disease.

**Funding:** We are very thankful to The University of Applied Sciences, Kaiserslautern, Germany and The Higher Education Commission Government of Pakistan (Ref. PD/OS-II/Batch-VI/Germany/2015/77159/17755) for providing financial support in the execution of the present project. These funding agencies provided support in term of stipend for authors and expenses concerning chemicals but did not have any additional role in the study design, data collection and analysis, decision to publish, or preparation of the manuscript. Additionally, Ursapharm Arzneimittel GmbH provided support in the form of salary for author PM, but did not have any additional role in the study design, data collection and analysis, decision to publish, or preparation of the manuscript. The specific roles of these authors are articulated in the 'author contributions' section.

**Competing interests:** The authors have read the journal's policy and have the following potential competing interests: PM is a paid employee of Ursapharm Arzneimittel GmbH. This does not alter our adherence to PLOS ONE policies on sharing data and materials. There are no patents, products in development or marketed products associated with this research to declare.

## Introduction

Neural stem cell (NSC) is a structural and functional unit of the nervous system [1] which deals with the traumatic events or neuronal losses with ageing. Being multipotent (capable of differentiating into glial cells, oligodendrocytes and neurons) in nature and having self-renewal properties [2], NSCs recapitulate the nervous system development processes such as proliferation, differentiation, migration, synaptogenesis and myelination [3, 4]. NSCs occur both in developing and adult nervous system of all mammalians including human [5]. Within the brain, NSCs are located mainly in the Subventricular zone (SVZ) and the dentate gyrus [6]. NSCs response to the external stimuli varies with the age of the donor [7], site in the nervous system and due to the diversity in their local environment [8, 9]. The SVZ presents the major niche of NSCs where primary and secondary neurogenesis occurs primarily [10–12].

The biggest hurdle in curing neurodegenerative disorders involves the irreversible damage to the neuronal cells which could no longer be replaced or repaired. High self-renewal potential, multipotency and multidirectional fate are a few rather unique characteristic features associated with NSCs which highlight the significance of these cells to serve as a promising tool to decipher the biochemical mechanisms underlying neurodegenerative disorders [13]. Screening small molecules which induce desired types of neurons from NSCs is highly valuable not only for regenerative medicine but also for the development of new drug candidates [14]. Interestingly, plant-based molecules, especially flavonoids, modulate the fate of NSCs favourably as confirmed by several *in vitro* culture systems. Enhanced proliferation of NSCs from multiple niches was observed when exposed to epimedium flavonoids [15], icariin [16, 17] and morin hydrate [18] whilst baicalin [19, 20], apigenin [21] and quercetin have proven their efficacy in inducing neuronal differentiation of NSCs [22]. Quercetin-3-o-glucuronide was reported to promote the cell migration of NSCs of mouse hippocampus [23]. It is important to note, however, that the bioavailability of flavonoids is generally low due to its inherent physicochemical properties.

Troxerutin (TRX) (3',4',7-tris[O-(2-hydroxyethyl)]rutin) is a water-soluble derivative of the bioflavonoid rutin extracted from the Japanese pagoda tree, which is also found abundantly in tea, coffee, vegetables and fruits. TRX exhibits several biological activities and cytoprotective effects against apoptosis, mitotic and necrotic cell death of liver, kidney and brain. TRX is also considered as an interesting drug candidate for multiple neurological disorders since it demonstrates antidepressant activity (because of its anti-inflammatory action), augments memory in animal models and provides anxiolytic actions (by reducing serum cortisol level) [24] and ameliorated the impairments of spatial learning and memory in a rat Alzheimer's model [25]. Moreover, TRX alleviates UV-B induced apoptosis, cell growth arrest, migration restriction, proliferation inhibition and DNA damaged in cultured HaCaT human immortal keratinocytes [26]. Intriguingly, TRX was reported to exhibit neuroprotective effects against the cholesterol-induced oxidative stress through its antioxidant properties and by enhancing phosphor inositide3 kinase/Akt activation in mouse hippocampus models. TRX exerts a neuroprotective role under endoplasmic reticulum induced stress by inhibiting the activities of caspase-3 and caspase-12. Overall, TRX is an excellent candidate which could be exploited to improve neuronal survival *i.e.* in Alzheimer's disease [27, 28].

Amyloid-ß (Aß) is a brain peptide with the size of approximately 4kDa derived from the amyloid precursor protein (APP) by enzymatic cleavage. Aß42 is the most hydrophobic form of Aß which is more fibrillogenic and forms the plaques in the brain [29]. Polymerization of monomeric Aß into soluble oligomer and insoluble fibril mass triggers Alzheimer's disease [30]. Predominantly, Aß deposition occurs in the hippocampus region of the brain which ultimately leads to the neuronal death due to oxidative stress. The production and aggregation of

Aß protein increase with aging [31]. Literature reveals that Aß42 inhibits proliferation and differentiation of NSCs from mouse hippocampus [32]. Flavonoids have proven their efficacy in improving synaptic functions against the Aß42 induced neurotoxicity [33]. TRX is capable of restoring memory loss and learning incapability induced by Aß42 in a rat model. These effects of TRX were associated with antioxidative action, anti-inflammatory effects and the capability to up-regulate cholinergic receptors in the animal brain [25]. Since neuroprotective and neuroaugmentation properties of TRX have already been reported in the literature, it would be interesting to decipher the effect of TRX on neurogenesis processes. So the aim of the present study is to establish an *in vitro* NSCs model from SVZ of the postnatal mice to investigate the effects of TRX on proliferation, migration and differentiation of NSCs and its neuroprotective effects against oligomeric Aß42 on the differentiation of NSCs.

## Materials and methods

### Animal dissection and cell culture

In the present study, NSCs were isolated from SVZ of postnatal Balb/c wild-type mice of 3–5 days old. Animals were housed under specific pathogen-free conditions on a 12 h light/12h dark cycle according to German regulations in the animal house facilities of the medical faculty Homburg, next to the Zweibrücken Campus. Animals were transported in warmed boxes and killed immediately after arrival by decapitation. Around three animals were employed used for each set of experiment. The total number of animals used for this study was 21. An authorized and well-trained researcher sacrificed the animals by decapitation without anaesthesia. Animal preparations in this study were carried out in strict accordance with the recommendation in the Guide for the care and use of laboratory animals according to animal protection law in Rhineland-Palatinate State, Germany. Since no experiment was directly performed on the living animals and only tissues were taken from the dead animals, no separate approval was necessary and the animal killing only has to be reported to the local Committee on the Ethics of Animal Experiments, University of Applied Sciences Kaiserslautern. NSCs were isolated and subsequently cultured according to the procedure reported in the literature with few necessary modifications [34]. Immediately after decapitation, mice brains were collected and stored in ice-chilled MEM-medium (Life technology, Eugene, USA) containing1% penicillin/streptomycin (Thermofischer, Waltham, USA). Under an inverted microscope (Olympus, Tokyo, Japan) SVZ was separated from both hemispheres followed by mechanical and enzymatic digestion with HyQtase enzymes (HyClone-GE, Utah, USA) and dissociated into single-cell suspensions. Approximately 100,000 cells were seeded in a proliferation medium DMEM/F12 (Life technology, Eugene, USA)containing 2% B-27 without antioxidants (Gibco, Paisley UK), 1% Penicillin/streptomycin, ß-mercaptoethanol, EGF 10 ng/mL and FGF 20 ng/mL (Immunotool, Friesoythe, Germany) in a T25 culture flask (Greiner, Frickenhausen, Germany). After 6 days NSCs proliferated to generate neurospheres. About half of the medium was replaced every alternative day. Before starting every individual experiment, neurospheres were dissociated and cell numbers were counted employing trypan blue (Gibco, Paisley, UK). All experiments were performed in five replicates (*n* = 5).

### Proliferation assay

The neurosphere clonogenic assay is a simple but robust assay which provides information about the effect of compounds on NSCs proliferation. Neurosphere's diameter and number were two readouts recorded at different time points of incubation [35]. Neurosphere diameter indicates NSCs multiplication within a neurosphere whilst neurosphere number is an indicator of self-renewal properties of NSCs [32]. At least 1000 cells were seeded in 200 μL of

proliferation medium into each well of a 96-well plate and medium was change on every alternative day. The cells were exposed to different concentrations of TRX (Y000497, Sigma-Aldrich, Taufkirchen, Germany) (25 μM, 50 μM and 100 μM). Neurosphere diameter and number were recorded on 3rd, 5th and 7th day of the culture. The effect of TRX was further confirmed by BrdU/Ki67 double staining. The ratio of BrdU-negative to the Ki67positivecells provides an accurate estimation of the amount of proliferating cells [36, 37]. NSCs cells were proliferated in proliferation medium supplemented by growth factors with and without TRX for 7 days. 4 h before the completion of incubation, proliferating cells were exposed to 10 μM BrdU (Sigma-Aldrich, Taufkirchen, Germany) followed by dissociation of neurospheres into a single-cell suspension. Around 20,000 cells were seeded on each of a 12 mm glass coverslips coated with PDL (Sigma-Aldrich, Taufkirchen, Germany) for 2 h. Cells were fixed with 4% Paraformaldehyde at 25°C for 20 min. BrdU/Ki67double staining was then performed as described in the immunostaining section. To confirm the stem cell nature of NSCs, NSCs after 7 days of proliferation were dissociated into a single cell suspension and seeded on ECM gel (Sigma-Aldrich, Taufkirchen, Germany) coated glass coverslips followed by fixation and stained for basic stem cell marker Nestin and a secondary NSCs marker GFAP. For calculating cell percentages, 30 microscopic fields at a magnification of 200 x were included in the observation.

## Differentiation assay

Effect of TRX on the differentiation of NSCs into neuronal and astrocyte cells was determined through a differentiation assay performed for 7 days and 48 h by exploiting all three test concentrations of TRX. Approximately 20,000 cells were attached on each of 12 mm glass coverslips coated with ECM gel. Cells were differentiated in the DMEM F12 medium comprising of a similar composition as employed for cell proliferation but excluding growth factors. NSCs in the differentiation medium were exposed to TRX for a given period followed by cell fixation and immunostaining for neuronal and astrocyte markers. Percentages of neurons and astrocytes were calculated and morphological parameters such as total neurite length, mean neurite length, %age of non-neurite neurons and soma area of astrocytes were quantified. For calculating cell percentages, 30 microscopic fields with a 20 x objective were captured and included in the observation. For morphological analysis, 100 cells were included in the analysis for each condition in each replicate. Morphological measurements were carried out for 7 days as well as for 48 h of incubation. Additionally, concentration produced optimum effect was used for counter toxicity testing against Amyloid-ß42 aggreSure AS72216 (AnaSpec EGT Group, California, USA) induced toxicity.

## Immunostaining

Adhered cells were fixed by exposure to 4% paraformaldehyde for 20 min at room temperature. After two times washing with PBS, cells were stored at 4°C until staining. Staining was performed following a recently published protocol [38]. Briefly, fixed cells were incubated with Triton 100 x 0.3% for 10 min at room temperature to enhance permeability followed by washing with PBS-tween one time and PBS 2 times. Non-specific binding was blocked with 10% normal donkey serum (Merck, Darmstadt, Germany) for 1 h at room temperature. Cultures were then incubated with primary antibodies mouse-anti-ß tubulin-IIIMAB1637 (1:500; Merck, Darmstadt, Germany), Nestin MAP353 (1:300; Merck, Darmstadt, Germany), rabbit anti-GFAP Z0334 (1:500, DakoGlostrup, Denmark), rat anti-BrdU (1:250; AbDserotec, Kidlington, UK) and rabbit-anti Ki67 (1:250; Abcam Cambridge, UK) at room temperature for 1hfollowed by washing 3 times with PBS. The samples were visualized using Alexafluor 488

(1:1000) and Alexafluor 594 (1:1000) conjugated donkey antibodies (Life Technology, Eugene USA) for another one hour and finally washing 3 times with PBS. DAPI was used for nuclear staining and incubated at room temperature for 10 min followed by final washing in PBS. Fluorescence supporting mounting medium was used to fix coverslips on glass slides. For BrdU, predenaturation of nucleic acid was achieved with 2N HCl prior to blocking and the acid was neutralized by borate buffer 0.1 M with pH 8.5 [39]. The rest of the steps were identical to those used previously.

## Neurosphere migration assay

NSCs were proliferated in proliferation medium supplemented by growth factor for 5–6 days in the absence of TRX. Around 12–15 neurospheres were allowed to attach on each of the 12 mm glass coverslips coated with PDL (Sigma-Aldrich, Taufkirchen, Germany) then incubated and differentiated in a differentiation medium excluding growth factor for a period of 24 hat 37˚C with 5% $CO_2$. The incubation was performed with and without TRX At the end of incubation, phase-contrast pictures were capture with a 4 x objective. Cell migration was evaluated by calculating the mean distance travelled by migrating cells away from the edge of the neurosphere in four directions measured at a right angle to the edge of the neurosphere core to the furthest migrated cells [40].

## Statistics

All results were presented as mean ± SEM which were calculated by exploiting descriptive statistics and non-parametric Kruskal-Wallis test with post hoc Dunn's test. Mann-Whitney pair wise test was employed in the case of two treatments.

# Results

## Identification of NSCs in vitro and the effects of TRX

We rated the stemness of NSCs culture using general stem cell markers Nestin and glial fibrillary acidic protein (GFAP), a reactive glial and astrocyte marker [41]. After 7 days of proliferation with and without TRX treatments, cells were fixed and immunostained. Our results indicated that each cell from every treatment condition was immunoreactive to a general NSCs marker Nestin. However, we also found a fraction of cells co-expressing Nestin and secondary NSCs marker GFAP in every treatment condition including control. TRX at 100 μM concentration significantly reduced the percentage of cells only expressing single Nestin marker when compared to all other treatments (Control vs 100 μM: *p = 0.046.*, 25 μM vs 100 μM: *p = 0.0018.*, 50 μM vs 100 μM: *p = 0.0043*) but increased the percentage of cells co-expressing Nestin and GFAP when compared with 25 μM (*p = 0.0013*) and 50 μM (*p = 0.0043*) concentrations (Figs 1 and 2D). Difference between Control and two lower concentrations of TRX was statistically non-significant concerning all calculated cell percentages.

## TRX effects on proliferation

We performed proliferation assay as previously reported that flavonoids enhance the proliferation of NSCs in an *in vitro* cultures [15, 16]. The results of clonogenic assay revealed that in the absence of growth factors, NSCs failed to proliferate and died on the 7[th] day in all tested concentrations except in Control+ve supplemented with growth factors where cells rapidly proliferated to form neurospheres at all observation time points (Fig 3). To investigate the augmenting effects of TRX on NSCs proliferation in the presence of growth factors, a set of a clonogenic assay where proliferation medium supplemented with growth factors for all

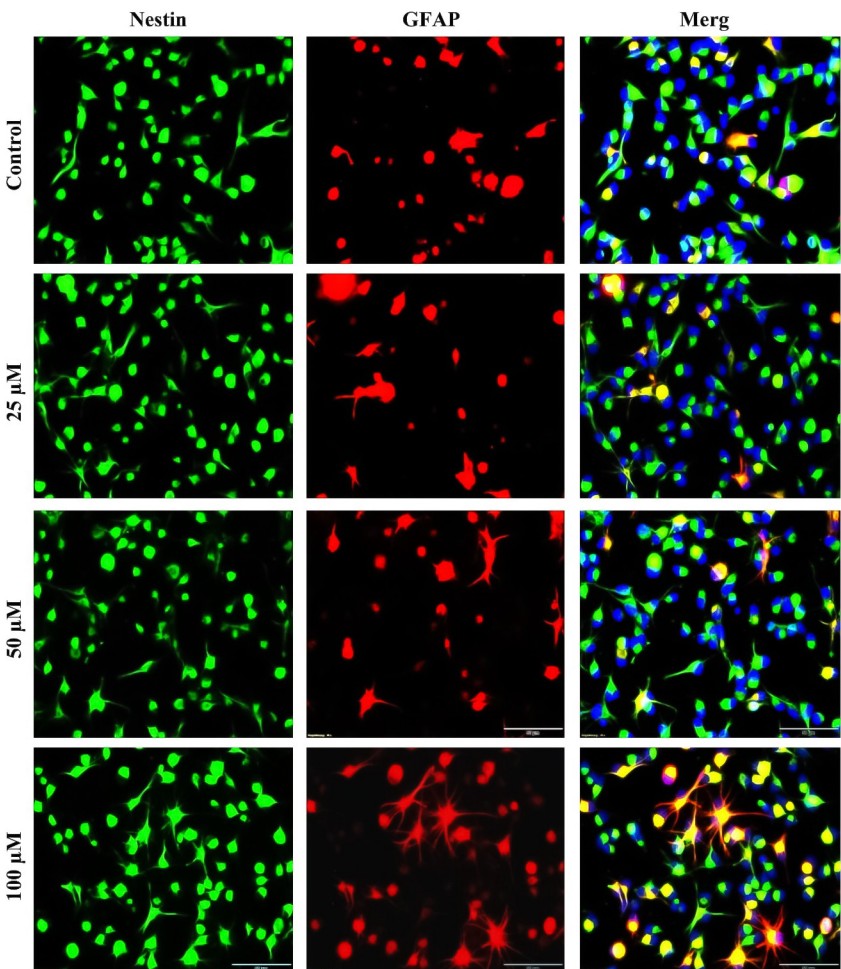

**Fig 1. Immunostaining of NSCs for stem cell markers after seven days of proliferation.** Green and red represent Nestin+ve and GFAP+ve cells, respectively. Yellow to orange cells in the merge images represent NSCs co-expressing both Nestin and GFAP. The highest concentration of TRX (100 μM) exhibited more GFAP and more doubled stained cells when compared to all other treatments. Pictures were captured with a 40 x objective of a fluorescent microscope. TRX; Troxerutin. Scale bar measures 50 μm.

treatment conditions was performed. The results clearly indicated that none of the tested TRX concentrations exhibited significant effect on NSCs proliferation parameters (neurosphere number and neurosphere mean diameter) during all observation points when compared to the Control (Fig 2A and 2B). The results of the study were further confirmed with more sensitive proliferation assay BrdU/Ki67 immunostaining. After seven days of proliferation, the effect of all tested concentrations of TRX on percentage of BrdU cells, Ki67 cells the ratio of BrdU/Ki67 were non-significant when compared to the Control (Fig 2C). Overall, the results demonstrate that TRX did not promote cell proliferation during any of the observation time points in the presence and absence of mitogenic growth factors.

## TRX enhanced neurite growth in neurons differentiated from NSCs

Being multipotent in nature, NSCs give rise to neurons, oligodendrocytes and glial cells on differentiation. NSCs differentiation mainly depends upon the environmental signals [42]. Plant

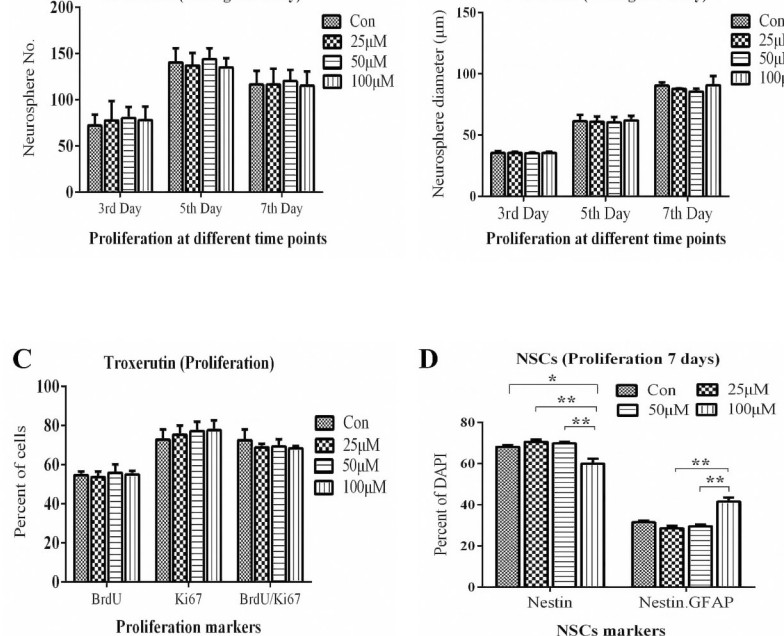

**Fig 2. TRX effect on neural stem cell markers and proliferation of NSCs.** TRX effect on proliferation was determined through neurosphere clonogenic and by BrdU/Ki67 double staining. **A**: Represents neurosphere number. **B**: Represents neurosphere mean diameter. **C**: represents the percentage of S-phase marker BrdU, Ki67 and the ratio of both markers. Percentage of BrdU+ve and Ki67 +ve cells were calculated from the total DAPI nuclei count. **D**: Represents the percentage of NSCs which were exclusively Nestin+ve and the fraction of NSCs co-expressing Nestin and GFAP. Every cell from each treatment condition was immunoreactive to general NSCs Nestin. Percentages of Nestin+ve and GFAP+ve cells were calculated from the total DAPI stained nuclei count. The experiments were performed as five replicates *(n = 5)*. Con; Vehicle control. Data were calculated as the mean ± SEM.*p<0.05,** p<0.01.

flavonoids are known to induce NSCs differentiation by interacting with genes regulating cell fate [20, 43–46]. In addition to neuronal differentiation, flavonoids also improve the neurite growth of differentiated neuronal cells [19]. Single-cell suspension of NSCs was differentiated for 7 days and 48 h in the presence and absence of TRX in 25 μM, 50 μM and 100 μM concentrations. Short term incubation was performed to evaluate the effect of TRX on early neuronal and astrocytes differentiation and also on their morphological characteristics. TRX increased the neuronal expression and decreased the GFAP astrocyte cells expression in a dose-dependent manner both on 7 days and 48 h of incubation. It is, however, important to mention that the results are statistically non-significant. Nevertheless, TRX in high concentration (*i.e.* 100 μM) exhibited a 15.9% increase in neuronal expression and an 11% decrease in astrocyte cells expression when compared to the vehicle Control. A similar trend was observed on 48 h incubation where the higher concentration increased neuronal expression by 19% with minimum effects on astrocyte expression (Fig 4A and 4B). TRX significantly decreased the percentage of non-neurite neurons (*i.e.* 2.2 folds compared to Control (*p = 0.0027*)) on 7th day at 100 μM (Figs 4C and 5). In the case of 48 h incubation, both 50 μM and 100 μM of TRX significantly reduced the percentage of non-neurite neurons *i.e.* 1.7 fold (*p = 0.05*) and 2.5 fold (*p = 0.0053*), respectively when compared to the Control. Additionally, 6–12% cells were double-stained for both neuronal and astrocyte markers during 48 h incubation (Figs 4D and 6). TRX reduced the percentage of double-stained cells in a dose-dependent manner and a

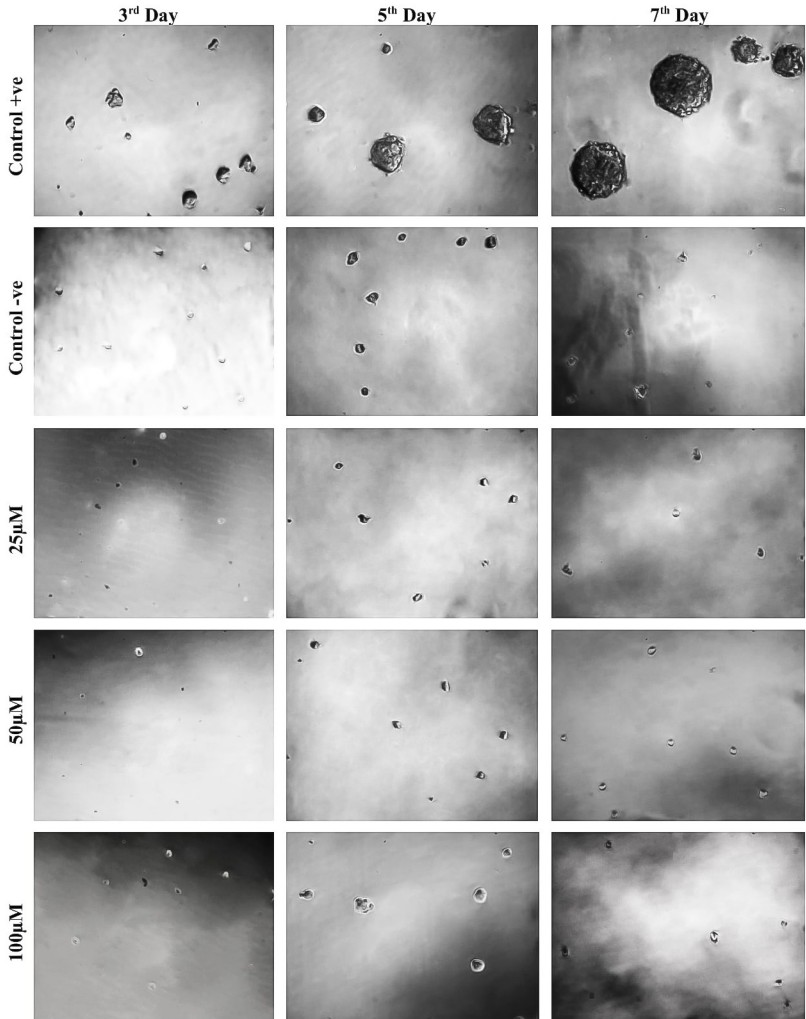

**Fig 3. Neurosphere clonogenic assay for NSCs proliferation.** The medium of Control+ve was supplemented with EGF and FGF growth factors. Growth factors were completely omitted from the Control-ve and all TRX treatments. Cells proliferated rapidly in positive control and formed neurospheres whose size increased with the increase in the incubation period. Cells in all other treatment conditions failed to proliferate and completely died on 7th day of observation. Pictures were captured with a 10 x objective of a phase-contrast microscope.

significant difference was observed for control vs 50 μM (*p = 0.044*) and Control vs 100 μM (*p = 0.01*) concentrations.

Total neurite length and mean neurite length per neuron were calculated as morphological parameters for neurons, while the cell body area was calculated for GFAP+ astrocytes. Morphological analysis was performed after 7 days employing an *in vitro* differentiation culture for all tested concentrations of TRX. The results revealed that TRX increased the total and mean neurite length in a dose-dependent manner after 7 days of incubation. Concerning total neurite length, a significant difference was observed between Control vs 50 μM (*p = 0.014*) and Control vs 100 μM (*p = 0.0009*) TRX stimulation (Figs 7A and 8). Neurite arborisation was assessed by calculating arborisation index as reported in the literature [47]. None of the tested concentrations of TRX affected the arborisation index after 7 days of differentiation. However,

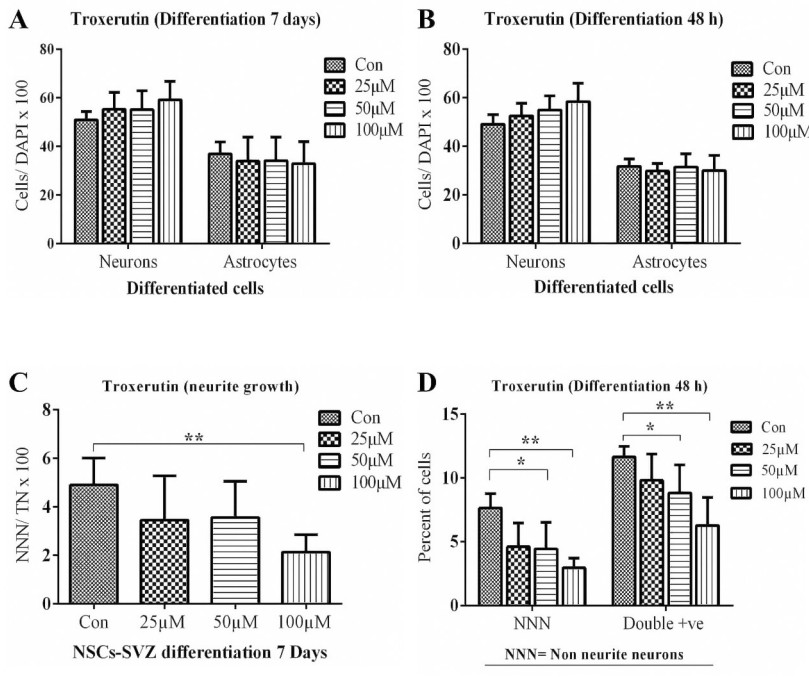

**Fig 4. TRX effect on NSCs differentiation after 7 days and 48 h of incubation.** ß-tubulin-III was employed as a neuronal marker and GFAP was exploited as an astrocyte marker. **A**: Represents the percentage of neurons and astrocytes after 7 days of differentiation exposed to different concentrations of TRX. **B**: represents the percentage of cells after 48 h of incubation. **C**: Represents the decrease in the percentage of non-neurite neurons; (NNN) with increasing concentration of TRX after 7 days. **D**: Represents the decrease in the percentage of non-neurite neurons and the double positive cells (Co-expressing ß-tubulin III and GFAP) with increasing concentrations of TRX after 48 h. NNN percentage was calculated from the total neuronal count. Only after 48 h of incubation time, cells immunoreactive to both neuronal and astrocyte markers (double+ve) could be detected. Data are presented as mean ± SEM. *p<0.05, ** p<0.01.

TRX significantly enhanced the number of neurite branching tips when compared to the Control (Fig 9) at both 50 µM *(p = 0.017)* and 100 µM *(p = 0.002)* concentrations. TRX increased the astrocyte soma area at 25 µM (*p = 0.05*) when compared to the Control and all other concentrations of TRX after 7 days of incubation (Figs 7C and 8). Morphological parameters of neurons and astrocytes for short term incubation of 48 h were only determined for high concentration of 100 µM because of optimum effects of this higher concentration on NSCs differentiation upon longer incubation. Interestingly TRX significantly increased total and mean neurite length and decreased the astrocyte's soma area at 100 µM when compared to the control (*p = 0.05*) (Figs 7B–7D and 10). Moreover, a non-significant difference concerning NSCs viability was also observed between the Control and 100 µM TRX Fig S A in S1 File).

## Neuroprotective effect of TRX against Aß42 induced depression in NSCs differentiation

Oligomeric form of Aß42 is implicated in Alzheimer's disease, a leading neurological disorder characterized by progressive loss of memory and cognitive functions [48, 49]. Oligomeric Aß42 decreases the neuronal and astrocytes differentiation of NSCs culture [32] whilst natural products, such as flavonoids including TRX, have proven their efficacy against the Aß42

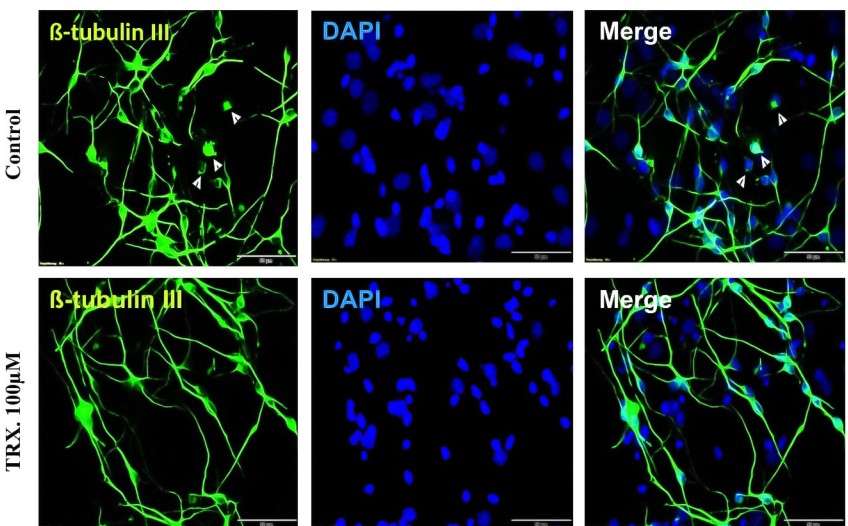

**Fig 5. Effects of TRX on NNN after 7 days of differentiation.** NSCs were differentiated for 7 days in the presence of TRX. Cells were fixed and immunostained for ß-tubulin III neuronal markers. Green cells represent neurons whilst blue rounded bodies indicate nuclei stained by nuclear dye DAPI. Control condition presented more neurons without neuritis (NNN) than the TRX as indicated by white arrows in the top left and top right images. Fluorescent images were captured with a 40 x objective of a microscope with scale bar measures 50 μm.

induced neurotoxicity both in *in vitro* and *in vivo* models [25, 49–51]. In this set of experiment, an *in vitro* model based upon NSCs isolated from SVZ of the developing mouse was established to unveil the neutralizing effect of TRX against the oligomeric Aß42 induced inhibition in the differentiation of NSCs and neurite growth. A concentration of 100 μM of TRX was selected as an optimal concentration to evaluate its neuroprotective effects against Aß42 induced inhibition in neuronal differentiation. Aß42 was employed in 10 μM concentration since this concentration has been reported to exhibit strong inhibitory effects on the neuronal differentiation [32] and the neurite growth in the differentiated neurons [52]. Moreover, the literature reveals that lower concentrations of Aß42 exhibit neurogenerative effects rather than neurotoxicity through a compensatory mechanism of brain repair [32, 53]. Finally, WST-1 cytotoxicity assay was performed to investigate the impact of Aß42 on the cell viability in differentiated culture for 48 h and a non-significant difference between the Control and the Aß42 (10 μM) was observed (Fig S B in S1 File). Aß42 significantly reduced the amount of neurons when compared to the Control (*p = 0.034*) and the Aß42+TRX combination (*P = 0.009*) after 48 h incubation (Fig 11A). The amount of non-neurite neurons significantly increased as compared to the Control (*p = 0.003*) and the combination Aß42+TRX (*p = 0.016*) in the remaining neurons. Aß42 treatment significantly increased the amount of cells co-expressing the neuronal and astrocytes markers when compared to the Control (*p = 0.005*) and Aß42+TRX combination (*P = 0.036*) (Figs 11B and 12). Aß42 reduced the total neurite length and mean neurite length of each neuron when compared with combination Aß42+TRX treatment (*p = 0.004*) (Figs 11C and 13) and also reduced total neurite length by 27% and mean neurite length by 19% when compared with the Control. Concerning the astrocyte soma area, both Aß42 (*p = 0.008*) and combination Aß42+TRX (*p = 0.007*) significantly reduced the soma area when compared to the Control (Figs 11D and 13). However, a non-significant difference was observed between Aß42 and Aß+TRX for this parameter.

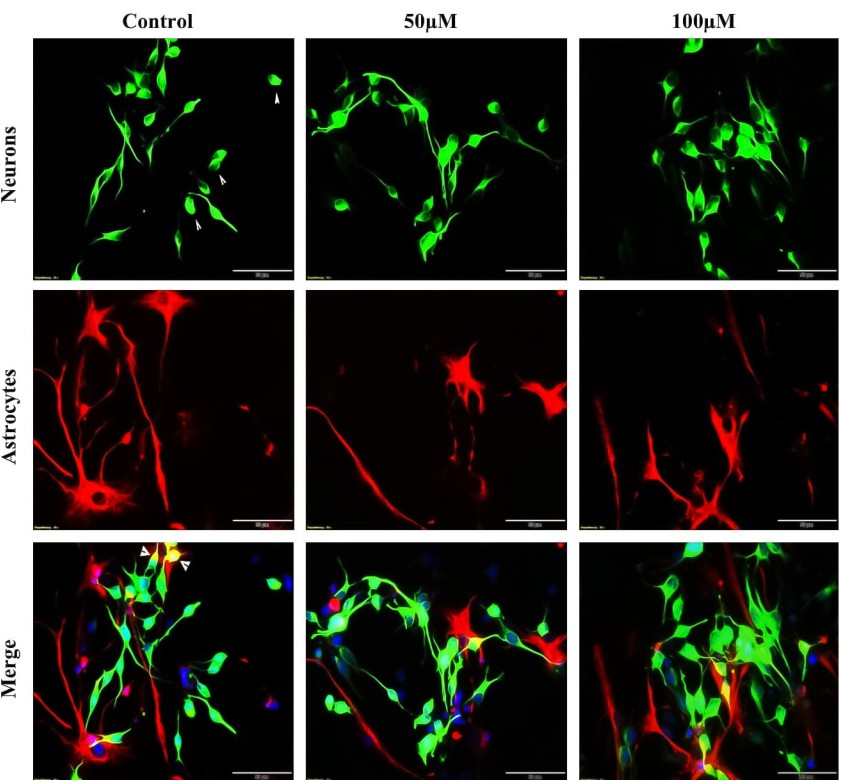

**Fig 6. Effects of TRX on NNN after 48 h of differentiation.** NSCs were differentiated for 48 h in the presence of TRX. Cells were fixed and immunostained for ß-tubulin III neuronal markers and GFAP astrocyte marker. Green cells represent neurons, red cells indicate astrocytes and blue rounded bodies present nuclei stained by nuclear dye DAPI. Control condition exhibited more neurons without neuritis (NNN) than TRX as indicated by white arrows in the top left image and more cells co-stained for neuronal and astrocyte (yellow cells) indicated by a white arrow in the bottom left images. Fluorescent images were captured with a 40 x objective of a microscope with scale bar 50 μm.

## TRX stimulates cell migration in NSCs culture

Stimulatory effects of TRX on cell migration have already been reported for different kinds of cells [26, 54]. TRX induces cell migration of human umbilical vein endothelial cells in combination with cerebroprotein hydrolysate [54]. To evaluate whether TRX influences migration in NSCs culture, neurosphere migration assays were performed. TRX significantly enhanced the cell migration at 50 μM (*p = 0.012*) and 100 μM (*p = 0.006*), while 25 μM of TRX did not exhibit any significant increase after 24 h incubation (Fig 14).

## Discussion

The self-renewal capacity and the differentiation into multiple cell types of the nervous system are characteristic features which make NSCs a very useful bio-tool for the screening of molecules which exert supporting effects on neural cell proliferation [15, 16, 55], differentiation [19, 20, 22], migration [23] and synaptogenesis [45]. Flavonoids serve as effective agents which provide antioxidative protection and neuroprotection. TRX is a derivate of rutin which serves as neuroprotective and neurogenerative agent [25, 28, 33, 56]. Additionally, TRX proved its effectiveness in avoiding Aß42 induced memory defects in a mouse model [27]. In the present study, an *in vitro* NSC based assay from SVZ of the postnatal mouse was established to evaluate

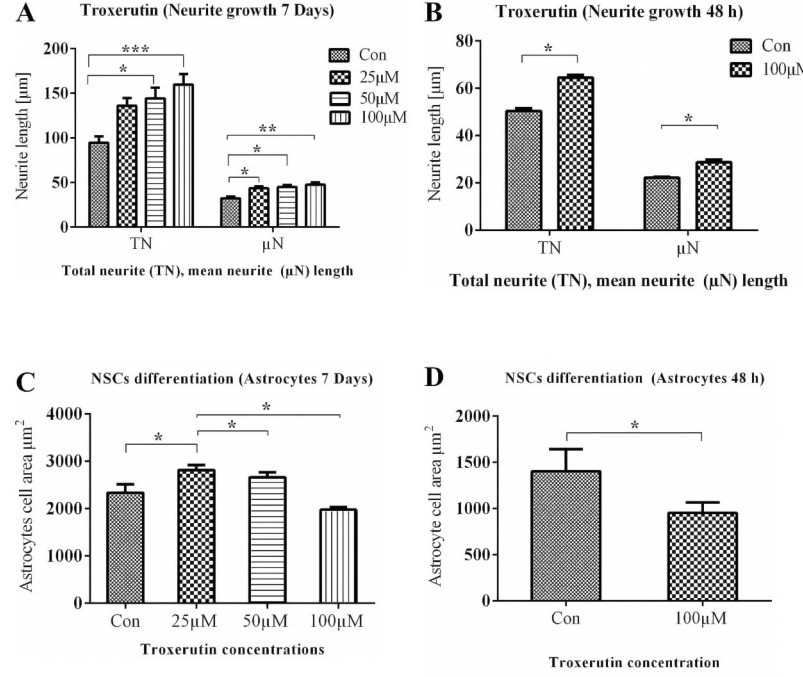

**Fig 7. Effects of TRX on morphological parameters of neurons and astrocytes differentiated from NSCs after 7 days and 48 h of incubation.** Approximately100 cells were employed for morphological analysis in each replicate for each condition. **A**: Represents TN and μN after 7 days. **B**: represents TN and μN after 48 h. **C**: Represents the astrocytes soma area after 7 days. **D**: represents astrocytes soma area after 48 h of incubation. The study was performed in 5 replicates *(n = 5)*. Data are presented as mean ± SEM. *p<0.05, **p<0.01, ***p<0.001. TN; total neurite length, μN; mean neurite length.

the effect of TRX on basic neurogenesis processes and to quantify neuroprotective effects in Aß42 challenges, where neuronal differentiation and neurite outgrowth are compromised.

Routinely, NSCs are identified in an *in vitro* culture by the immunostaining of NSCs/ neural progenitor cells (NPCs) markers namely DCX, Atoh1, SOX2, Nestin and GFAP. DCX is a marker for NPCs and not pure NSCs. Atoh1 is the marker for NSCs of the Cochlar nucleus. SOX2 is the marker of NSCs in the developing brain during early embryonic phase. Nestin is the widely employed marker for both NSCs and NPCs in the developing and adult nervous system [57–59]. Nestin is an intermediate filamentous protein widely expressed by NSCs from the mammalian nervous system [15]. There exists a class of NSCs which co-express nestin and the reactive glial cell marker GFAP. So both Nestin and GFAP are considered as NSCs markers [60]. NSCs which co-express Nestin and GFAP comprise of radial glial cells which present neural stem cell properties [61, 62]. These cells predominantly exist in the SVZ of both developing and adult brain and are capable of differentiation into neurons, oligodendrocytes and astrocytes *in vitro* [63, 64]. In this study, both nestin and GFAP immunostainings were performed to identify NSCs isolated from SVZ culture proliferated for 7 days in the presence of different concentrations of TRX. The immunostaining results indicated that cells treated with TRX (100 μM) presented a tendency to increase the percentage of cells co-expressing both stem cell markers Nestin and GFAP when compared to all other treatments. It is already reported in the literature that around 30% of the total NSCs isolated from mammalian brain stained for stem cell marker nestin, also co-stained for astrocyte marker GFAP [41]. These

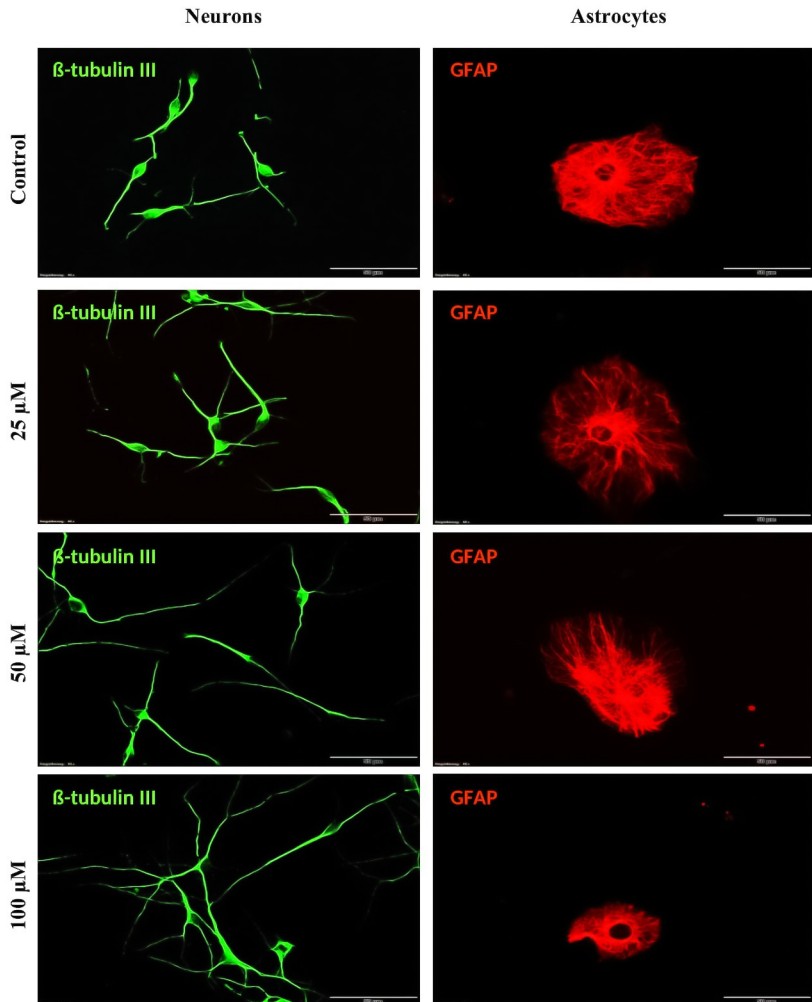

**Fig 8. Effect of TRX on neurite growth of neurons and soma area of astrocyte after 7 days of incubation.** NSCs were differentiated on ECM coated glass coverslips fixed and then stained for a neuronal marker (ß-tubulin-III) and astrocyte marker (GFAP). Green cells represent are neurons. Neurons under TRX treatments exhibited elongated neurites with more branching points when compared to control. Red cells indicate astrocytes. Most of the astrocytes cells presented leaf-like morphology. Astrocytes treated with TRX in less than 25 µM concentration demonstrated expended soma area when compared with all other treatment conditions. Cells presented a smooth surface with no deformation. All measurements were performed exploiting Cell SENE software. Pictures were captured with 40 x objective of a fluorescent microscope with scale bar 50 µm.

doubled stained NSCs give rise to neurons and glial cells at the time of birth and give rise to adult NSCs in the SVZ thus allowing a continuous supply of NSCs for regenerative processes throughout the life [65]. Additionally GFAP expressing NSCs from SVZ quickly turn into functional neurons in response to brain injury [66]. Another similar property of NSCs is their ability to facilitate neuronal cell migration [67]. Taken together, our immunostaining results suggested that all cells cultured from mouse SVZ were NSCs as they expressed stem cell markers in proliferation culture and then differentiated into neurons and astrocytes on subsequent differentiation experiments.

The proliferation of NSCs in an *in vitro* culture condition can only be maintained when cells are supported by essential growth factors. Among these, brain-derived neurotrophic

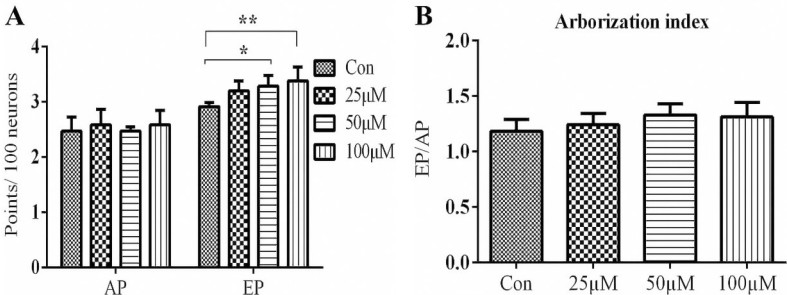

**Fig 9. Effect of TRX on neurite arborisation after 7 days of differentiation.** Neurite arborisation measurements were carried out for NSCs culture differentiated for 7 days with different concentrations of TRX. Around 100 neurons were included in the observation for each condition in each replicate. **A**: Represents the effects of TRX on neurite attachment point; AP and neurite ending points; EP. AP is an indicator of primary neurite number and EP is an indicator of neurite branching. **B**: Represents the neurite arborisation index which is described as the ratio of EP to that of AP. Data are presented as mean ± SEM. *p≤0.05.

factors (BDNF), fibroblast growth factor (FGF) [68] and epidermal growth factor (EGF) are very important [69]. In the absence of growth factors, such as EGF and FGF, NSCs failed to proliferate and died within a few days of culture. The literature reveals that various natural flavonoids facilitate the maintenance of the *in vitro* proliferation of NSCs independent of the

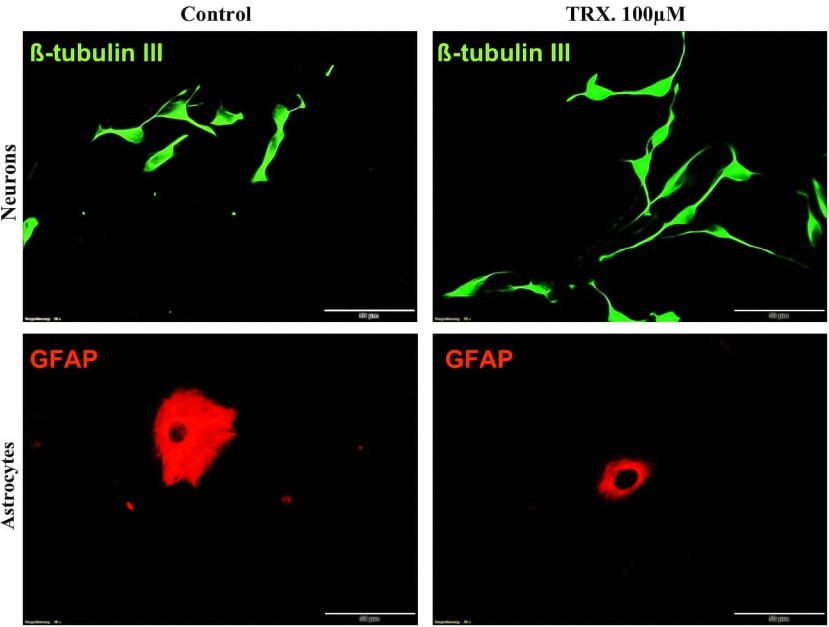

**Fig 10. Effect of TRX on neurite growth of neurons and soma area of astrocyte after 48 h of incubation.** NSCs differentiated on ECM coated glass coverslip for 48 h of incubation. Cells were immunostained for a neuronal marker (ß-tubulin III) and an astrocyte marker (GFAP). Green cells represent neurons. Neurons treated with TRX present elongated neurites with more branching points when compared to control. Red cells represent astrocyte cells stained for GFAP. Astrocytes presented both leaf-like and star-like morphology but leaf-like morphology was dominant over star shaped cells. Astrocyte cells treated with The Control exhibited larger soma area when compared to cell treated with 100 μM of TRX. Cells presented a smooth surface with no deformation. All measurements were performed exploiting Cell-SENE software. Pictures were captured with a 40 x objective with 50 μm scale bar.

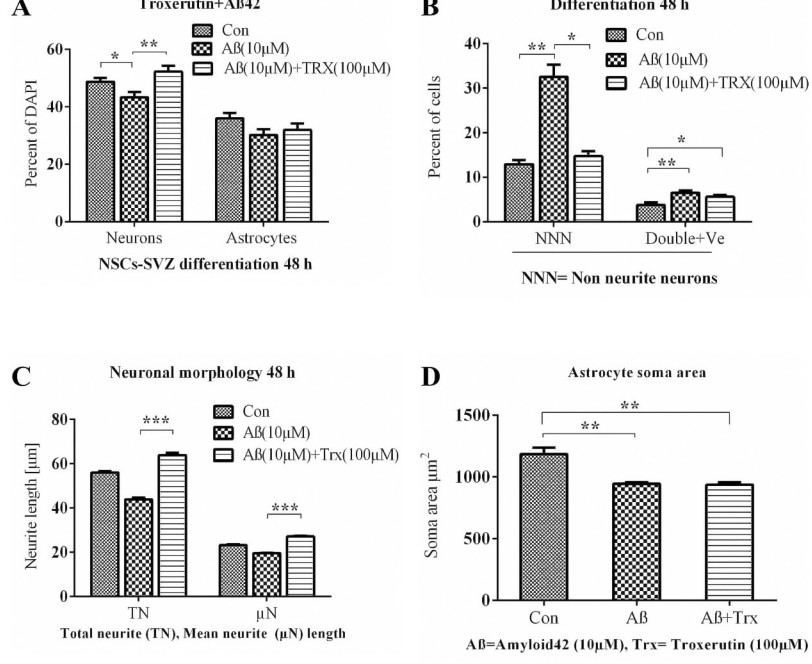

**Fig 11. Neuroprotective effects of TRX flavonoid against Aß42 induced depression of differentiation after 48 h of differentiation on ECM coated glass coverslips. A**: Represents the percentage of cells differentiated from NSCs. **B**: represents the percentage of non neurite neurons differentiated from total neurons and the percentage of cells double +ve for both neuronal marker ß-tubulin III and astrocyte marker (GFAP). **C** and **D** represent the effects on neurite outgrowth per neuron and astrocyte soma area, respectively. The data were conceived exploiting Cell SENE software. Percentage of differentiated neurons and astrocytes were calculated from the total DAPI nuclei count. At least 100 cells were included in observation for morphological analysis for each condition in each replicate. Data are presented as mean ± SEM. Experiments were performed as 5 replicates (n = 5).*p<0.05, **p<0.01, ***p<0.001.

growth factors. Epimedium flavonoids, for instance, promote the proliferation of cultured NSCs from postnatal rat hippocampus [15]. Icariin has been reported to promote the proliferation of NSCs by up-regulating cell cycle gene D1 and protein p21 [16]. Results of our clonogenic and BrdU/KI67 immunostaining assays clearly indicated that none of the tested TRX concentrations exhibited stimulatory or inhibitory effects on the proliferation of NSCs irrespective of the presence or absence of essential neurotrophic factors. Previous studies reported a stimulatory effect of TRX on cell proliferation other than NSCs. 10 μM concentration of TRX, for instance, alleviated UV induced arrest in the proliferation of cultured human keratinocytes cell lines (HaCaT) by upregulating miRNA -181a-5p [26]. Moreover, TRX in combination with cerebroprotein lysate enhanced the proliferation of human umbilical vein endothelial cells HUVECs [54].

Being multipotent in nature, NSCs give rise to functional cells of the nervous system on differentiation in response to environmental stimuli [15]. In this study NSCs were differentiated in the presence of different concentrations of TRX for 48 h and for a relatively long period of 7 days. Purpose of short term differentiation was to evaluate the effect of TRX on early neuronal differentiation and neurite outgrowth. One week differentiation of NSCs provided neurons with well-developed neurite and branching pattern. Higher concentrations of TRX stimulated the neurite outgrowth and decreased the percentage of non-neurite neurons when compared to the vehicle control during both incubation periods. Effect of TRX on neurite arborisation

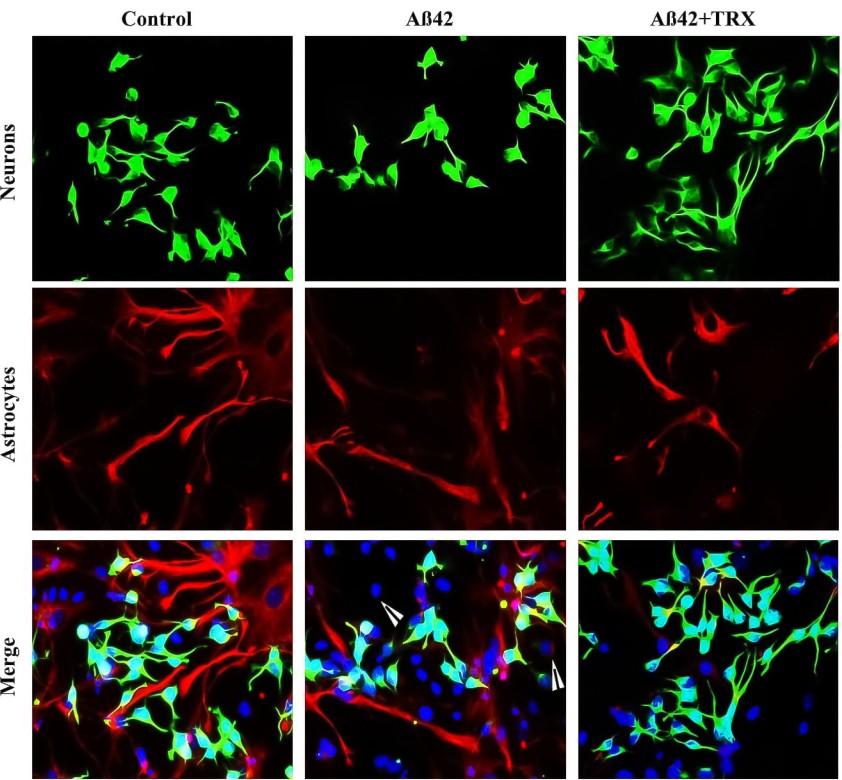

**Fig 12. Neuroprotective effect of TRX against Aß42 induced inhibition in the amount of neurons and astrocytes after 48 h of differentiation.** Green cells present ß-tubulin III neurons which were found denser in control and Aß42 +TRX. Red cells represent GFAP astrocytes which were observed in both leaf-like and star-like morphology. Astrocytes were relatively denser in Control as compared to other treatment conditions. Pictures were captured with a 40 x objective of a fluorescent microscope. Scale bar is 50 μm.

index was non-significant at lower concentration but the higher concentrations of 50 μM and 100 μM significantly enhanced the number of neurite branching tips. Although there are no data available to represent the effect of TRX on the differentiation of NSCs, several studies describe the neurogenerative effects of flavonoids. Baicalin, for instance, enhanced the neuronal fate of cultured NSCs isolated from rat hippocampus [46] and also enhanced the neurite outgrowth by upregulating phosphorylation of Erk1/2 [19]. Baicalin also stimulated the neuronal differentiation and inhibited glial differentiation of rat embryonic NSCs by modulating the function of transcription factor stat3 and basic helix-loop-helix gene family [20]. Quercetin is another flavonoid which enhanced neurogenesis and synaptogenesis by stimulating brain-derived neurotrophic factors and phosphorylating cyclic AMP response binding protein (pCRBP) [22]. In another study quercetin enhanced neurite outgrowth and affected the percentage of neuronal cells by upregulating Gap-43 and cAMP in a cultured N1E115 cell line [44]. Prenylated flavonoid ENDF1 enhanced the axonal length and branching density in the cultured neurons from the dorsal ganglion by upregulating the expression of microtubule binding-protein gene DCX and maintaining the $Ca^{2+}$ haemostasis in neurons [70]. Oral administration of flavonoids rich dried root extracts of Chinese herb *Scutellaria baicalensis* Georgi stimulated the axonal growth against experimentally induced spinal injury in a rat model by upregulating the expression of NF-H expression in neurons [71]. Isoquercitrin

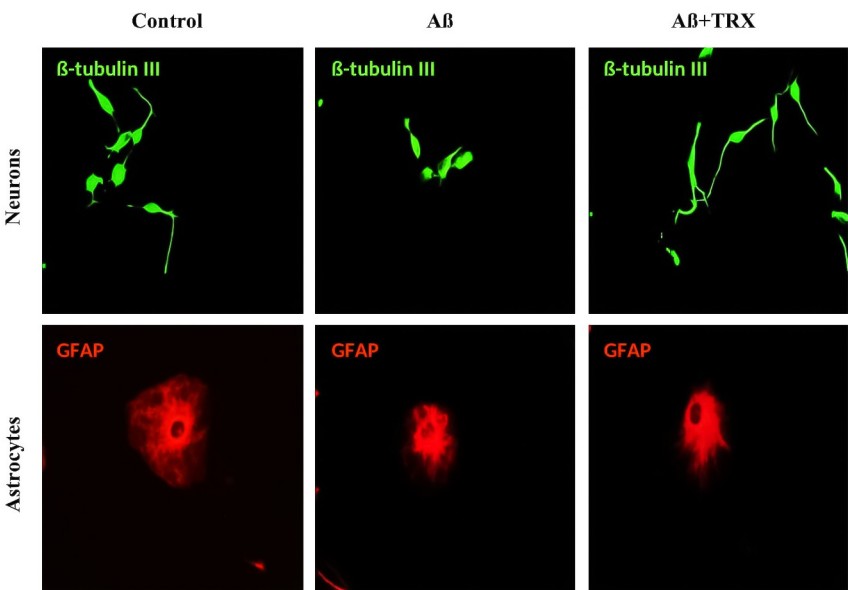

**Fig 13. Neuroprotective effect of TRX against Aß42 induced neurite growth inhibition after 48 h incubation.**
Green cells represent ß-tubulin III neurons. Neurons treated with TRX+Aß42 exhibited elongated neurites with more branching when compared with control. Red cells represent GFAP astrocyte which under control condition exhibited larger soma area when compared with Aß and combination TRX+Aß42 treatment. Cells demonstrated a smooth surface with no deformation. All measurements were performed by employing Cell SENE software. Pictures were captured with a 40 x objective with 50 μm scale bar.

flavonoids promoted the axonal elongation in of the cultured NG108-15 cells by reducing the activity of RhoA kinase [45]. A high concentration of TRX also decreased the percentage of double positive cells (immunoreactive to both neuronal and astrocyte markers) which was only observed during 48 h of incubation. The occurrence of cells co-expressing neuronal markers ß-tubulin III and glial markers GFAP in SVZ of the developing brain is supported by the previously reported literature. These cells are neural progenitors and on long term differentiation, give rise to either neurons or glial cells [72, 73]. In this study, 100μM TRX presented a tendency to reduce the percentage of ß-tubulin/GFAP double-positive cells and, at the same time, increased the percentage of neurons.

TRX in high concentration (100 μM) significantly reduced astrocyte soma area as compared to control and two other lower concentrations. The literature reveals that TRX exhibits a mitigating effect in Parkinson's 6-OHDA rat model not only *via* antioxidation activity but also through inhibition of astroglial GFAP expression partially by modulating the function of PI3K/ERβsignalling pathway [56]. In another study, baicalin inhibited astrocyte differentiation of rat embryonic NSCs by interacting with basic helix-loop-helix genes and transcription factor stat3 [20]. Reduced oxidative stress stimulates the neuronal differentiation of NSCs [74]. TRX exhibits antioxidant activities by interacting with reactive oxygen species [75]. Since TRX exerts neuroprotective and neurogenerative effects through its anti-oxidative actions in the brain tissues of rat [54], it was assumed that anti-oxidation activity might also be associated with enhanced neuronal differentiation and neurite growth. Detailed anti-oxidation and molecular studies are required to prove this assumption.

Aß42 decreased the percentage of neurons, decreased the neurite outgrowth and neurite length after 48 h of incubation. Literature reveals that Aß42 decreased the percentage of

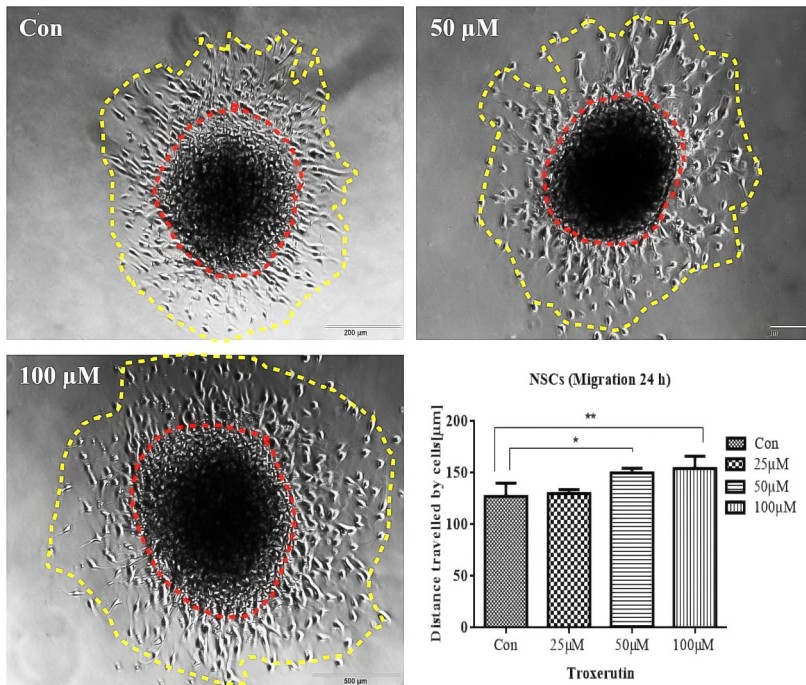

**Fig 14. TRX enhances the migration of differentiated cells from NSCs cultured for 24 h.** NSCs were proliferated to develop neurospheres which were adhered to PDL coated glass coverslips and incubated in differentiation mediumwith and without the presence of TRX for 24 h. The central dark area encircled by The red dotted ring is the neurospheres core consists of the heterogeneous population of cells. Results representmean distance travelled by cells from the core in all four directions (**Graph**). The migrated cells are enclosed in yellow dotted ring. For each condition, 10-15neurospheres were included in the observation. The study was performed in 5 replicates *(n = 5)*. Pictures were captured with 10 x objective of a phase-contrast microscope. Data are presented as the mean ± SEM.

neurons differentiated from the cultured mouse NSCs isolated from the hippocampus [32]. Another *in vitro* study reported that Aß42, on short term incubation, significantly inhibited the axonal growth and synapsis formation in cultured cortical and hippocampus cells. These inhibitory effects of Aß42 are similar to those defined in transgenic mouse and Alzheimer pathology [76]. Moreover, short term exposure to Aß42 inhbits neurite growth of cultured PC12 cells due to an oxidative stress and mitochondrial dysfunction as described by researchers [52]. TRX, in high concentration (*i.e.*100 μM), neutralized the inhibitory effects of Aß42 (10 μM) on neuronal differentiation, neurite outgrowth, neurite extension after 48 h incubation. TRX protects hippocampus neurons from the neurotoxic effects of Aß42 by ameliorating antioxidant enzymes and attenuating elevated acetylcholinesterase enzyme levels. Additionally TRX also reduced the apoptosis on chronic treatment of 14 days [33]. Apigenin is an aglycone flavone which improves the memory defect induced by Aß25-35 in mouse by several mechanisms including antioxidant actions, reduction of acetylcholinesterase activity and modulationof phosphor-CREB, BNDF and TrkB [77]. Quercetin has been reported to exhibita protective effect against Aß42 induced lipid peroxidation in cultured hippocampus cell culture from postnatal rat [78].

Countertoxicity effects of a flavonoid against Aß 42 depend upon the ability of a flavonoid to prevent the fibrillization of the later [49]. Two major structural requirements for the anti-fibrilization effect of a flavonoid molecule involve the number of aromatic rings and the

number of hydroxyl groups present in the molecule. Aromatic rings bind with the hydrophobic amino acid residues through covalent bonds and hydroxyl groups of a flavonoid interact with hydrophilic amino acid residues of the peptide backbone of Aß 42 leading to the disaggregation effects or prevention of fibrillization. The number of these functional groups are, therefore, directly proportional to the antifibrils activity against Aß 42. Gallocatechin gallate and theaflavin exhibited 100% efficacy in preventing fibrils formation of Aß 42 in an extracellular chemical reaction since these molecules provide a higher number of aromatic rings and hydroxyl groups in their structures as compared to other compounds tested (Phan 2019). A number of these aromatic rings and reactive hydroxyl groups in TRX molecule [79] is comparable to that of theaflavin [80], rutin [81] and more than that of gallocatechin gallate [82] so it was assumed that the neutralizing activity of TRX against Aß 42 induced inhibition in neuronal differentiation and neurite growth in our experiments might be due to the disaggregating or antifibrillization effects of TRX. Future detailed studies are needed to prove if TRX prevents fibrillization of Aß42 in a reaction mixture.

The findings of this study indicated that TRX can stimulate cell migration in NSCs culture in the absence of any stress. Ma et al., reported in their study that TRX in combination with cerebroprotein hydrolysate, induced cell migration of human umbilical vein endothelial cells [54]. In another study, TRX induced HaCa cells migration by ameliorating UV induced migration restriction [26]. Stimulatory effects of TRX on the migration of cell lines are mediated through modulation of regulatory gene miR-181a-5p and transcription factor integrin ß3 mRNA [26, 54]. Quercetin has been reported to induce murine NSCs migration under differentiation conditions with concomitant up-regulation of CXCR4 gene in an *in vitro* experiment [23]. CXCR4 is a receptor protein for chemokine SDF1. CXCR4/SDF1 signalling pathway which is implicated in the development of various tissues including the nervous system. One of the major roles of SDF1/CXCR4 involves the regulation of neuronal cell migration of various kinds in cortex and cerebellum areas of the brain. Knock out of either CXCR4 or SDF1 genes in mouse resulted in severe defects in granular cell migration [83].

## Conclusions

The present study revealed that TRX not only promoted the neuronal differentiation of cultured NSCs of the postnatal mouse but also stimulated neurite outgrowth and neurite extension as well as cell migration in the absence of any inhibitory stimuli. Moreover, TRX also neutralized the inhibitory effects of Aß42 oligomer on neuronal differentiation and neurite outgrowth. These findings provide clues about the role of TRX in neurogenesis and curing Aß42 dependent neurological disorders. TRX, in stark contrast to unmodified rutin and other flavonoids, is easily water soluble, which makes this unique molecule a suitable candidate for oral applications. Indeed, detailed investigations are required to explore molecular pathways governing stimulatory effects of TRX in neurite growth, cell migration and neuroprotective actions against neurotoxic peptides implicated in neurodegenerative disorders.

## Supporting information

**S1 File.**
(DOCX)

## Acknowledgments

We are grateful to Anne Braun, Anna Baumann and Steven Schulte for technical support in laboratory-based work and Dr. Abdul Baseer for administrative work.

## Author Contributions

**Conceptualization:** Muhammad Irfan Masood, Karl Herbert Schäfer.

**Data curation:** Muhammad Irfan Masood, Mahrukh Naseem, Maximilian Weyland, Peter Meiser.

**Formal analysis:** Karl Herbert Schäfer.

**Funding acquisition:** Karl Herbert Schäfer.

**Investigation:** Muhammad Irfan Masood, Karl Herbert Schäfer.

**Methodology:** Muhammad Irfan Masood.

**Project administration:** Karl Herbert Schäfer.

**Resources:** Karl Herbert Schäfer, Peter Meiser.

**Software:** Muhammad Irfan Masood, Mahrukh Naseem, Maximilian Weyland.

**Supervision:** Karl Herbert Schäfer.

**Visualization:** Karl Herbert Schäfer.

**Writing – original draft:** Muhammad Irfan Masood, Mahrukh Naseem, Maximilian Weyland, Peter Meiser.

**Writing – review & editing:** Muhammad Irfan Masood, Mahrukh Naseem, Maximilian Weyland, Peter Meiser.

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
