## [Decision Letter · Decision Letter 0]

1 May 2020

PONE-D-20-07816

Troxerutin Flavonoid has Neuroprotective properties and increases Neurite outgrowth and Migration of Neural stem cells from the Subventricular zone

PLOS ONE

Dear Dr. Masood,

Thank you for submitting your manuscript to PLOS ONE. After careful consideration, we feel that it has merit but does not fully meet PLOS ONE’s publication criteria as it currently stands. Therefore, we invite you to submit a revised version of the manuscript that addresses the points raised during the review process.

Specifically, I share the opinion of reviewer#1 that (i) your cell culture system has to be better characterized to conclusively demonstrate it is NSC, (ii) figure quality has to improve, (iii) Ab42 use has to be better rationalized, (iv) potential extracellular effects of your drug on Ab42 toxicity should be controlled, (v) all statements such as those about axons have to be supported by data, (vi) effects on NSC differentiation should be shown (after all that is the ultimate proof of preservation of NSC stemness), (vii) relevant literature should be cited. I am willing to reconsider your submission only if all those points are adequately addressed.   

We would appreciate receiving your revised manuscript in 4 months after receipt of this decision letter. To enhance the reproducibility of your results, we recommend that if applicable you deposit your laboratory protocols in protocols.io, where a protocol can be assigned its own identifier (DOI) such that it can be cited independently in the future. For instructions see: http://journals.plos.org/plosone/s/submission-guidelines#loc-laboratory-protocols

We look forward to receiving your revised manuscript.

Kind regards,

Michal Hetman

Academic Editor

PLOS ONE

Journal Requirements:

'In the present study Animal preparations were carried out following the guidelines of the institutional ethics committee (University of applied sciences Kaiserslautern) and according to animal protection law in Rhineland-Palatinate state, Germany.'

a. Please amend your current ethics statement to confirm that your named ethics committee specifically approved this study.

For additional information about PLOS ONE submissions requirements for ethics oversight of animal work, please refer to http://journals.plos.org/plosone/s/submission-guidelines#loc-animal-research.

'We are very thankful to The University of Applied Sciences, Kaiserslautern, Germany and The Higher Education Commission Government of Pakistan for providing financial support in the execution of the present project.'

We note that one or more of the authors are employed by a commercial company: URSAPHARM Arzneimittel GmbH

4. Please add the concentration or dilution factor of all antibodies used to your Methods section.

5. To comply with PLOS ONE submissions requirements, in your Methods section, please provide additional information on the animal research and ensure you have included details on

(i) the source and number of mice,

(ii) whether anesthesia was used prior to decapitation and, if so, your methods of anesthesia and/or analgesia, and

(iii) whether decapitation was done by trained personnel.

7. PLOS requires an ORCID iD for the corresponding author in Editorial Manager on papers submitted after December 6th, 2016. Please ensure that you have an ORCID iD and that it is validated in Editorial Manager. To do this, go to ‘Update my Information’ (in the upper left-hand corner of the main menu), and click on the Fetch/Validate link next to the ORCID field. This will take you to the ORCID site and allow you to create a new iD or authenticate a pre-existing iD in Editorial Manager. Please see the following video for instructions on linking an ORCID iD to your Editorial Manager account: https://www.youtube.com/watch?v=_xcclfuvtxQ

Reviewers' comments:

Reviewer's Responses to Questions

**Comments to the Author**

1. Is the manuscript technically sound, and do the data support the conclusions?

Reviewer #1: No

Reviewer #2: Yes

2. Has the statistical analysis been performed appropriately and rigorously? 

Reviewer #1: I Don't Know

Reviewer #2: Yes

3. Have the authors made all data underlying the findings in their manuscript fully available?

Reviewer #1: Yes

Reviewer #2: Yes

4. Is the manuscript presented in an intelligible fashion and written in standard English?

Reviewer #1: No

Reviewer #2: Yes

5. Review Comments to the Author

Reviewer #1: The has aspects that are technically well done, but lacks critical controls needed to sup[port several of the conclusions.

Major Issues:

1) Figure 1 "stemness" is determined by staining with nestin and GFAP. This is insufficient to examine the developmental state of the dispersed NSCs (cells). For example Sergent-Tanguay eyt al (2006) J Neurosci Res find that astroglial cells can be nestin+/GFAP+. Additional studies should include more rigorous characterization of the cells used in the work. This is critical information needed to interprete all other studies.

2) The neurosphere data in Fig. 3 is unconvincing. Only a single neurosphere is shown (and in many cases these images are out of focus). A representative field including many spheres should be shown.

3) For Fig. 4, it does not appear that the impact of TRX on NSC differentiation is ever examined. Cultures would need to be temporally characterized for the presence of NSC, immature/mature neurons, oligodendrocytes, and glial numbers - this should be done and the percentage of each cell type included in the analysis. It is also unclear what NNN cells are - could these be dead or dying neurons? Other cell types? Images are needed to support these studies.

4) The authors indicate TRX-dependent changes in axonal branch points- but no data is supplied to support this statement.

5) It is unclear why AB42 was used to explore TRX-mediated survival. Is it know where TRX exposure alters AB42 oligmerization/ability to form fibrils? Could what is considered cell survival be a reduction in AB42 in vitro toxicity? Does TRX treatment cause pro-survival molecular changes (described in paper below)? Are the GFAP+/nestin+ cell simply astrocytes. Finally, astrocytes play a role in amyloid production - might differences in astrocye number impact these studies? This study mirrors a published study: Farajdokht et al. 2017 PMID 29285004. This work should be acknowledged and included in the Discussion.

Overall:

The image quality is very poor for the majority of figures, making the review significantly more difficult. The manuscript would be improved by careful review and editing my a native English speaker.

Reviewer #2: Overall the paper is easily read, but there are still some minor problems in language and spelling. A throughout checking is recommended

a previous study has shown that behavior of stem cells depend on physiochemical properties of materials and drug ( https://doi.org/10.1002/admi.201900572). please discuss your article in keeping with this article and explain which property of troxerutin flavonoid resulted in differentiation of cells?

6. PLOS authors have the option to publish the peer review history of their article (what does this mean?). If published, this will include your full peer review and any attached files.

Reviewer #1: No

Reviewer #2: No

---

## [Author Response · Author response to Decision Letter 0]

23 Jun 2020

The Detailed response to the Reviewer#1, Reviewer#2 and the Editor comments are given in the attachment " Response to Reviewer" of this submission system along with new references and Figures.

---

## [Decision Letter · Decision Letter 1]

20 Jul 2020

Troxerutin Flavonoid has Neuroprotective properties and increases Neurite outgrowth and Migration of Neural stem cells from the Subventricular zone

PONE-D-20-07816R1

Dear Dr. Masood,

We’re pleased to inform you that your manuscript has been judged scientifically suitable for publication and will be formally accepted for publication once it meets all outstanding technical requirements.

Kind regards,

Michal Hetman

Academic Editor

PLOS ONE

Additional Editor Comments (optional):

Reviewers' comments:

Reviewer's Responses to Questions

**Comments to the Author**

1. If the authors have adequately addressed your comments raised in a previous round of review and you feel that this manuscript is now acceptable for publication, you may indicate that here to bypass the “Comments to the Author” section, enter your conflict of interest statement in the “Confidential to Editor” section, and submit your "Accept" recommendation.

Reviewer #1: All comments have been addressed

Reviewer #2: All comments have been addressed

2. Is the manuscript technically sound, and do the data support the conclusions?

Reviewer #1: Partly

Reviewer #2: Yes

3. Has the statistical analysis been performed appropriately and rigorously? 

Reviewer #1: Yes

Reviewer #2: Yes

4. Have the authors made all data underlying the findings in their manuscript fully available?

Reviewer #1: Yes

Reviewer #2: Yes

5. Is the manuscript presented in an intelligible fashion and written in standard English?

Reviewer #1: Yes

Reviewer #2: Yes

6. Review Comments to the Author

Reviewer #1: The authors have addressed my earlier concerns. I urge them to review the article once more as there remain grammatical issues, but these are not viewed as an impediment to publication.

Reviewer #2: ------------------ My recommendation is accept. ----------------------------------------------------------------------------------------------------------------------------------------------------------------------------

7. PLOS authors have the option to publish the peer review history of their article (what does this mean?). If published, this will include your full peer review and any attached files.

Reviewer #1: No

Reviewer #2: No

---

## [Editor Report · Acceptance letter]

31 Jul 2020

PONE-D-20-07816R1 

Troxerutin Flavonoid has Neuroprotective properties and increases Neurite outgrowth and Migration of Neural stem cells from the Subventricular zone 

Dear Dr. Masood:

I'm pleased to inform you that your manuscript has been deemed suitable for publication in PLOS ONE. Congratulations! Your manuscript is now with our production department. 

Kind regards, 

on behalf of

Dr. Michal Hetman 

Academic Editor

PLOS ONE